# The function of a heterozygous *p53* mutation in a Li-Fraumeni syndrome patient

Yang Li[1☉], Ting Li[1☉], Yuejia Tang[1☉], Zhiyan Zhan[1], Lixia Ding[1], Lili Song[1], Tingting Yu[2], Yi Yang[1], Jing Ma[3], Yingwen Zhang[1], Ying Zhou[4], Song Gu[5], Min Xu[5]*, Yijin Gao[1]*, Yanxin Li[1]*

**1** Department of Hematology & Oncology, Key Laboratory of Pediatric Hematology and Oncology Ministry of Health, Shanghai Children's Medical Center, Shanghai Jiao Tong University School of Medicine, Shanghai, China, **2** Molecular Biological Diagnostic Laboratory, Shanghai Children's Medical Center, Shanghai Jiao Tong University School of Medicine, Shanghai, China, **3** Department of Pathology, Shanghai Children's Medical Center, Shanghai Jiao Tong University School of Medicine, Shanghai, China, **4** Department of Radiology, Shanghai Children's Medical Center, Shanghai Jiao Tong University School of Medicine, Shanghai, China, **5** Department of General Surgery/Surgical Oncology Center, Shanghai Children's Medical Center, Shanghai Jiao Tong University School of Medicine, Shanghai, China

☉ These authors contributed equally to this work.
* liyanxin@scmc.com.cn (YL); gaoyijin@scmc.com.cn (YG); xumin@scmc.com.cn (MX)

**Data Availability Statement:** All relevant data are within the paper and its Supporting Information files.

**Funding:** This work was supported in part by the National Key R&D Program of China

## Abstract

p53 is one of the most extensively studied proteins in cancer research. Mutations in *p53* generally abolish normal p53 function, and some mutants can gain new oncogenic functions. However, the mechanisms underlying *p53* mutation-driven cancer remains to be elucidated. Our study investigated the function of a heterozygous *p53* mutation (p. Asn268Glufs*4) in a Li-Fraumeni syndrome (LFS) patient. We used episomal technology to perform somatic reprogramming, and used molecular and cell biology methods to determine the *p53* mutation levels in patient-originated induced pluripotent stem (iPS) cells at the RNA and protein levels. We found that p53 protein expression was not increased in this patient's somatic cells compared with those of a healthy control. *p53* mutation facilitates the proliferation of tumor cells by inhibiting apoptosis and promoting cell division. It can inhibit the efficiency of somatic reprogramming by inhibiting OCT4 expression during reprogramming stage. Moreover, not all *p53* mutant iPS cell lines have mutant p53 RNA sequences. A small percentage of mutant p53 mRNA is present in the somatic cells from the patient and his mother. In summary, this *p53* mutation can promote tumor cell proliferation, inhibit somatic reprogramming, and exhibit random *p53* allelic expression of heterozygous mutations in the patient and iPS cells which may be one of the reasons why the people with *p53* mutations develop cancer at random. This finding suggested that mutant *p53* allelic expression should be added to the risk forecasting of cancer.

## Introduction

Somatic cell reprogramming is a valuable tool for understanding the mechanism of pluripotency recovery, because it enables the possibility of producing patient-specific pluripotent

(2018YFC1313000/2018YFC1313005 Y. L and Y. G); the National Natural Science Foundation of China (No.81470315 and No. 81972341 to Y. L.); the Shanghai Jiao Tong University Medical Engineering Cross Fund (No. YG2017MS32); the Local High Level University Construction Project of Shanghai Jiao Tong University School of Medicine; the Commission of Shanghai Municipality (No. 17441903200 to S. G and No. 17411950402 to M. X.); and the Pudong New Area Science & Technology Development Fund (PKJ2018-Y47) to Y. L.

**Competing interests:** The authors have declared that no competing interests exist.

stem cells [1–3]. What's more, researchers can get infinite patient samples and set up experimental platforms to study the pathogenesis of diseases in vitro [4].

As a tumor suppressor gene, p53 plays a significant role in promoting apoptosis and cell cycles arrest. Missense mutations of p53 can be a key factor of cell carcinogenesis and reduce the induction efficiency of induced pluripotent stem cells (iPS) [5–12]. Moreover, the p53 mutation might not only loss its anti-cancer functions, but also obtain oncogenic traits called gain of function (GOF), including malignant progression and invasion, metastasis and even chemotherapy resistance [13–16]. In cell reprogramming, oncogenes, such as Notch, can inhibit the generation of iPS cells [17], but no one knows how specific *p53* mutations affect the iPS cell derivation process. Additionally, p53 does not fully follow the classic Knudson's two-hit theory during carcinogenesis or cancer progression [18].Therefore, so many healthy people with the same *p53* mutation can go their entire lives without developing cancer [9].

In the present study, we generated iPS cells from the peripheral blood of a male infant with LFS; the patient has a *p53* heterozygous mutation inherited from his mother (22 years old) [19]. The p53 mutation facilitates the proliferation of tumor cells by inhibiting apoptosis and promoting cell division. Additionally, it reduced the reprogramming efficiency by inhibiting Oct4 expression. In three mutant *p53* iPS cell lines, we found that the expression levels of WT p53 protein in one iPS line was different from that in the other two iPS cell lines. We speculated that the differential expression of WT p53 was related to allelic expression imbalance. Using p53 RNA sequencing, we confirmed this conclusion.

## Materials and methods

### Cell culture

Primary murine embryonic fibroblasts (MEFs) with *p53* knockout were obtained from 13.5-day CD-1 IGS mouse embryos. HEK293T and MEF cells were cultured in standard DMEM containing 10% FBS (HyClone, Logan) and passaged routinely with trypsin-EDTA solution. Human iPSCs were maintained in a feeder-free culture system. Briefly, the wells of plates were precoated with Matrigel (BD Biosciences), and then we seeded the iPSCs and cultured them in PSCeasy medium (Cellapy).

### Isolation and preparation of MNCs from peripheral blood

Blood samples were obtained from the Hematology and Oncology Department of Shanghai Children's Medical Center, and patient's mother provided informed consent. MNCs were isolated from PB samples using standard Ficoll procedures; 8 ml of diluted blood (blood: PBS = 1:2) was loaded onto a 3 ml layer of Ficoll-Paque PREMIUM (p = 1.077 g/ml; Sigma) in a 15-ml conical tube.

### Culture and expansion of MNCs from peripheral blood

We expanded PB MNCs for 4–10 days in a serum-free medium supplemented with a mixture of cytokines. We used erythroid culture medium (ECM). ECM included IMDM (50%; Invitrogen) and Ham's F12 (50%; Invitrogen) with ITS-X (100×; Invitrogen), chemically defined lipid concentrate (100×; Invitrogen), L-glutamine (100×; Invitrogen), BSA (5 mg/ml; Sigma), ascorbic acid (0.05 mg/ml; Sigma), L-thioglycerol (200 µM; Sigma), IL-3 (10 ng/ml; PeproTech), SCF (100 ng/ml; PeproTech), erythropoietin (2 U/ml; PeproTech), dexamethasone (1 µM; Sigma), IGF-1 (40 ng/ml; PeproTech), and holotransferrin (100 µg/ml; R&D).

## Nucleofection and generation of iPSCs

The following episomal vectors were used: pEV SFFV-OCT4-E2A-SOX2 (OS), pEV SFFV-MYC-E2A-KLF4 (MK), and pEV SFFV-BCL-XL (Bcl-XL). We added plasmids (4 μg OS (EF1-OS), 4 μg MK (EF1-MK) and 2 μg B (BCL-XL)) to a sterile Eppendorf tube and mixed them with 100 μl nucleofection buffer (Nucleofector™ Kits for Human CD34+ Cells, Lonza) and then transferred the mix to the cell pellet ($1 \times 10^6$ cells). Using the plastic pipette provided by the kit, we transferred the mixture of plasmids and cells into the provided cuvette to run the program (U008) for nucleofection (2B; Lonza). After nucleofection, we directly transferred the mixture to a culture plate, which was already preseeded with feeder cells. The cells were then cultured in reprogramming medium, which was composed of knockout DMEM/F12 medium (Invitrogen) supplemented with 1% L-glutamine (Invitrogen), 2 mM nonessential amino acids (Invitrogen), 1% penicillin/streptomycin (Invitrogen), 50 ng/ml FGF2 (Invitrogen), 1% ITS (BD Biosciences), and 50 μg/ml ascorbic acid (Sigma) for 7 days. The cells were then cultured in E8 medium (Invitrogen) until iPSCs were generated.

## Generation of mouse iPS cells

Retroviral constructs pMXs-Klf4 (#13370), pMXs-Sox2 (#13367), pMXs-Oct4 (#13366), pMXs-c-Myc (#13375) [1], were obtained from Addgene. Reprogramming of primary (passage 2) MEFs was performed as previously described [12]. In brief, primary MEFs of the indicated genotypes were seeded in 100-mm-diameter dishes ($5 \times 10^5$ cells per dish) that had been pre-coated with 0.1% gelatin (Sigma). They were transduced twice in the next two days at 24 h intervals by virus supernatant collected from Plat-E cells transfected with the previously mentioned retroviral plasmids. At the end of transduction, we changed the medium to mouse ES culture medium. After culturing for 10–12 days, colonies with ES-cell-like morphology became visible. They were then chosen after counting or picking for further expansion on feeder fibroblasts using standard ES culture methods.

## Counting and picking of iPSC colonies

When the colonies became visible to the naked eye, we stained the human iPS cells with a Tra-1-60 antibody, counted them under a fluorescence microscope and picked them by hand. To pick them, we gently scratched a colony with a 10 μl pipette tip and transferred the single colony to a 12-well plate coated with Matrigel and filled with E8 medium. We usually selected 10 to 20 colonies from each donor. Mouse iPS cell colonies were counted using a published method [12].

## Cell line construction

WT *p53*, mutant *p53* p.Asn268Glufs*4 and *p53* R175H coding DNA sequences (CDS) were cloned into a pLL CMV puro mammalian lentiviral expression vector.

 To produce the lentivirus, each expression vector was transfected into 293T cells with second-generation lentiviral packaging plasmids pMD2.G and psPAX2 using the PolyExpress transfection reagent (Excellgen, Rockville). Forty-eight and 72 h after transfection, we harvested the culture medium, incubated it with Lenti-X concentrator (Clontech Laboratories, Mountain View), and centrifuged it to obtain concentrated lentivirus. *p53*$^{-/-}$ MEF cells were infected with the lentiviruses in the presence of 6 μg/mL polybrene (Sigma-Aldrich) for 24 h. Overexpression was confirmed by Western blotting.

## Immunohistochemistry

IHC was performed in 3-μm formalin-fixed paraffin embedded tissue sections mounted on adhesive microscope slides. Sections were deparaffinized, rehydrated in graded alcohols and underwent antigen retrieval performed by microwave treatment in 0.01 M-citrate buffer at pH 6.0, during 9 min. The sections were then incubated overnight at 4˚C with the primary antibody against p53 (1:100, monoclonal antibody; Cat. MAB-0674; MXB). The detection of the immune reaction was performed using the avidin-biotin-peroxidase method (1:100; Vector Laboratories, Peterborough, UK). DAB (3, 3′-diaminobenzidine) was used as chromogen and hematoxylin as nuclear counterstaining.

## Immunofluorescence

To detect targeted antigens and p53 in pluripotent stem cells, we immobilized cells with PBS containing 4% polyformaldehyde at room temperature for 10 minutes. After washing with PBS, the cells were incubated in PBS containing 0.1% Triton X-100 for 20 minutes at room temperature. Then, we stained fixed cells with SSEA-4 (1:100; monoclonal antibody; MAB8490; Stemgent), TRA-1-60 (1/200; monoclonal antibody; 09–0010; Stemgent), OCT4 (1/200; monoclonal antibody; MAB4419A4; Millipore), Nanog (1/600; monoclonal antibody; sc-293121; Santa Cruz) and p53 (1/500; monoclonal antibody; ab1101; abcam). These primary antibodies were visualized by with goat anti-rabbit IgG bound to Alexa 488 and goat anti-rabbit IgG bound to Alexa 594 or goat anti-mouse IgG bound to Alexa Fluor 488. Nuclear staining was performed with DAPI, and fluorescence images were obtained using Zeiss inverted LSM confocal microscopy (Carl Zeiss).

## Teratoma formation assay and histological analysis

We suspended the human iPSCs in PBS at $1 \times 10^8$ cells/ml and then injected 100 ml of cell suspension ($1 \times 10^7$ cells) subcutaneously into the dorsal side of SCID mice. One month after the injection, we dissected the tumors from the mice. Teratomas were weighed and fixed in PBS containing 4% formaldehyde and embedded in paraffin wax. We then produced sections from the fixed teratomas and stained them with hematoxylin and eosin.

## Gene expression analysis of LFS iPS cell lines

We used three LFS iPS cell lines, as well as H1 ESCs and H9 ESCs, and we extracted total RNA from each using the RNeasy plus kit (Qiagen) to assess their self-renewal abilities and *p53* transcription levels. Real-time PCR was performed using the SYBR Green PCR Master Mix (Applied Biosystems) on a 7500 Fast Real-Time PCR System (Applied Biosystems). The primer sets were as follows: Oct4, 5′-ATTCAGCCAAACGACCATCT-3′ and 5′-GCTTCCTCCACCCA CTTCT-3′; SOX2, 5′-CACACTGCCCCTCTCACAC A-3′ and 5′-CCCTCCCATTTCCCTCGT TT-3′; NANOG, 5′-GCCGAAGAATAGCAATGGTGTG-3′ and 5′-GGAAGATAGAGGCTG GG GTAG-3′. p53, 5′-CTGAGGCATAACTGCACCCT-3′ and 5′-GACAA GGGTGGTTGGGAGTA G-3′.

To determine the average copy numbers of residual or integrated episomal vectors in iPSC clones, real-time PCR analysis was performed. We extracted total DNA (genomic and episomal) from iPSCs at passage 10. Two sets of primers were used to detect vector DNA (in either the episomal or integrated form): EBNA1, 5′-TTTAATACGATTGAGGGCGTCT-3′ and 5′-GGTTTTGAAGGATGCGATTAAG-3′; and OSW, 5′-GGATTACAAGGATGACGACGA-3′ and 5′-AAGCCATACGGGAAGCAATA-3′.

## Gene expression analysis of MEF and iPS cell lines

To detect the expression of pluripotent genes in MEF cells at different time points of reprogramming, we collected SSEA1-positive cells and extracted total RNA from the groups of *p53* mutant, WT, or an empty vector control for 2, 4, 8 and 12 days using a RNeasy plus kit (Qiagen). Real-time PCR was performed using the SYBR Green PCR Master Mix (Applied Biosystems) on a 7500 Fast Real-Time PCR System (Applied Biosystems). The primer sets were as follows: OCT4, 5′- TCTTTCCACCAGGCCCCCGGCTC-3′ and 5′- TGCGGGCGGACATGGGG AGATCC-3′; SOX2, 5′-TTGCCTTAAACAAGACCACGAAA-3′ and 5′- TAGAGCTAGACTCC GGGCGATGA-3′; NANOG, 5′- CAGGTGTTTGAGGGT AGCTC-3′ and 5′- CGGTTCATCA TGGTACAGTC-3′; GAPDH, 5′- TGTGTCCGTCGTGGATCTGA-3′ and 5′ TTGCTGTTGA AGTCGCAGGAG-3′.

## Growth curve

Cell growth curves were compared among the cells of *p53* mutant, WT, or an empty vector control according to the method [20]. Briefly, 1.5 E5 cells were seeded in a 12-well plate, and the growth curves were plotted by counting cells every 24 hours over three-day with excel software.

## Karyotyping and G-banding

G-banding chromosome analysis of the iPSC lines was performed following the protocol published by Li et al [12]. A certified cytogenetic technologist interpreted the data.

## Western blotting

Cell extracts were prepared, resolved on gels, transferred to nitrocellulose and incubated with antibodies against the N terminus of p53, which can recognize mutant and wild type of p53 (1:1,000; monoclonal antibody; ab1101; abcam), β-actin (1:500; monoclonal antibody; M1210-2; Huaan), BCL-2 (1:1,000; monoclonal antibody; sc-7382; Santa Cruz), and PUMA (1:500; monoclonal antibody; sc-374223; Santa Cruz). γH2AX-139 (1:1,000; monoclonal antibody; sc-517348; Santa Cruz).

## Apoptosis

Apoptosis was measured by staining with annexin V–APC and Propidium Iodide (PI)-phycoerythrin (PE) (Annexin V-APC Apoptosis Detection kit, BD Pharmingen) followed by flow cytometry on a FACS flow cytometer (BD, Canto II). All experiments were performed in triplicate, and results were calculated as the mean ± S.D.

## Statistical analysis

All experiments were repeated three times. Data are presented as the mean ± S.D. Two-tailed Student's *t* tests were performed, and $p < 0.05$ was considered statistically significant.

## Animals and ethics statement

SCID mice were bought from Shanghai SLAC Laboratory Animal CO. All mice used in this study were authorized by the Animal Care Use and Review committee of Shanghai Jiao Tong University. The study was conducted according to the Ethical Principles of Measures for Ethical Review of Biomedical Research Involving Human Beings and the Declaration of Helsinki.

The ethics committee of the Children's Medical Center affiliated with Shanghai Jiao tong University approved the induction experiment for iPS cells (SCMCIRB-K2014050).

## Results

### *p53 Asn268Glufs*\*4 mutation was found in a LFS patient

The tumor suppressor gene *p53* encodes a tetrameric DNA-binding protein that regulates cell cycle and apoptosis [21–23]. A 6-month-old male infant was first diagnosed with composite ACC (adrenocortical carcinoma) and neuroblastoma in May 2017. In March 2018, the relapse of ACC was identified by abdominal computed tomography (CT) scanning and confirmed by resection (Fig 1A and 1B). We found that p53 protein expression was negative in this patient's adrenocortical carcinoma tumor tissues by immunohistochemistry (IHC) (Fig 1C). Since gain-of-function mutations of *p53* were reported to be stable for IHC [24–26], our data suggest that this new *p53* mutation is not a gain-of-function mutation. Given that p53 gene mutation has a strong correlation with the diagnosis of infant ACC [19, 27], genetic testing for p53 status was performed on the patient and his parents with their agreements. We found a heterozygous insertion of c.801dupG that caused a p.Asn268Glufs\*4 in the *p53* gene in this patient and his mother, suggesting that this patient inherited the mutation from his healthy mother (Fig 1D). The active p53 is a homo-tetramer formed by four identical chains of 393 residues each, and the N-terminal region of p53 consists of an intrinsically disordered transactivation domain (TAD) and a proline-rich region. It is followed by the central, folded DNA-binding core domain that is responsible for sequence-specific DNA binding. Via a flexible linker, this domain is connected to a short tetramerization domain that regulates the oligomerization state of p53 (Fig 1E). *Asn268Glufs*\*4 mutation* is a nonsense mutation which is located in specific DNA binding domain, which caused early termination of this specific protein synthesis and may affect the function of p53.

### p.Asn268Glufs\*4 mutation of p53 loses some functions of wild type p53

To explore the function of the mutant *p53*, we separately infected lentivirus-mediated *p53* mutant, wild type (WT), or an empty vector (EV) control into *p53⁻/⁻* MEF. We identified full-length WT and the mutant (truncated) of p53 protein in the transformed cells (Fig 2A). As p53 known functional mutant R175H, the expression of BCL-2 in mutant cells was similar to that in control cells, and higher than that in *p53* WT cells. The expression of PUMA in mutant cells was the same as that in the control cells, but PUMA levels relatively increased in WT cells (Fig 2A, S1C Fig). The analysis of apoptosis revealed that compared with unregulated control cells, overexpression of WT p53 enhanced apoptosis in *p53⁻/⁻* MEF cells (Fig 2B, S1E Fig). Only p53 WT induced DNA damage compared with the control (Fig 2D and 2E, S1C Fig). Unlike p53 R175H, *p53 p.Asn268Glufs*\*4 mutant as well as its WT dramatically inhibited cell proliferation (Fig 2C, S1D Fig). These data suggest that *p53* mutant lost WT p53 ability to induce apoptosis and DNA damage and thereby reduced the inhibition of cell division.

### The *p53 p.Asn268Glufs*\*4 mutation inhibits iPS cell generation

Since p53 is critical for iPSC reprogramming [6, 10, 12], we explored the role of this p53 mutant in iPSC reprogramming. We first tested the expression of p53 protein in mononuclear cells from the patient and his mother. As shown in Fig 3A, the patient showed lower p53 WT protein levels compared with his mother and the healthy control and a truncated p53 protein was found only in the patient's MNC, suggesting that the p53 mutant protein does not express in all cells with this gene mutation.

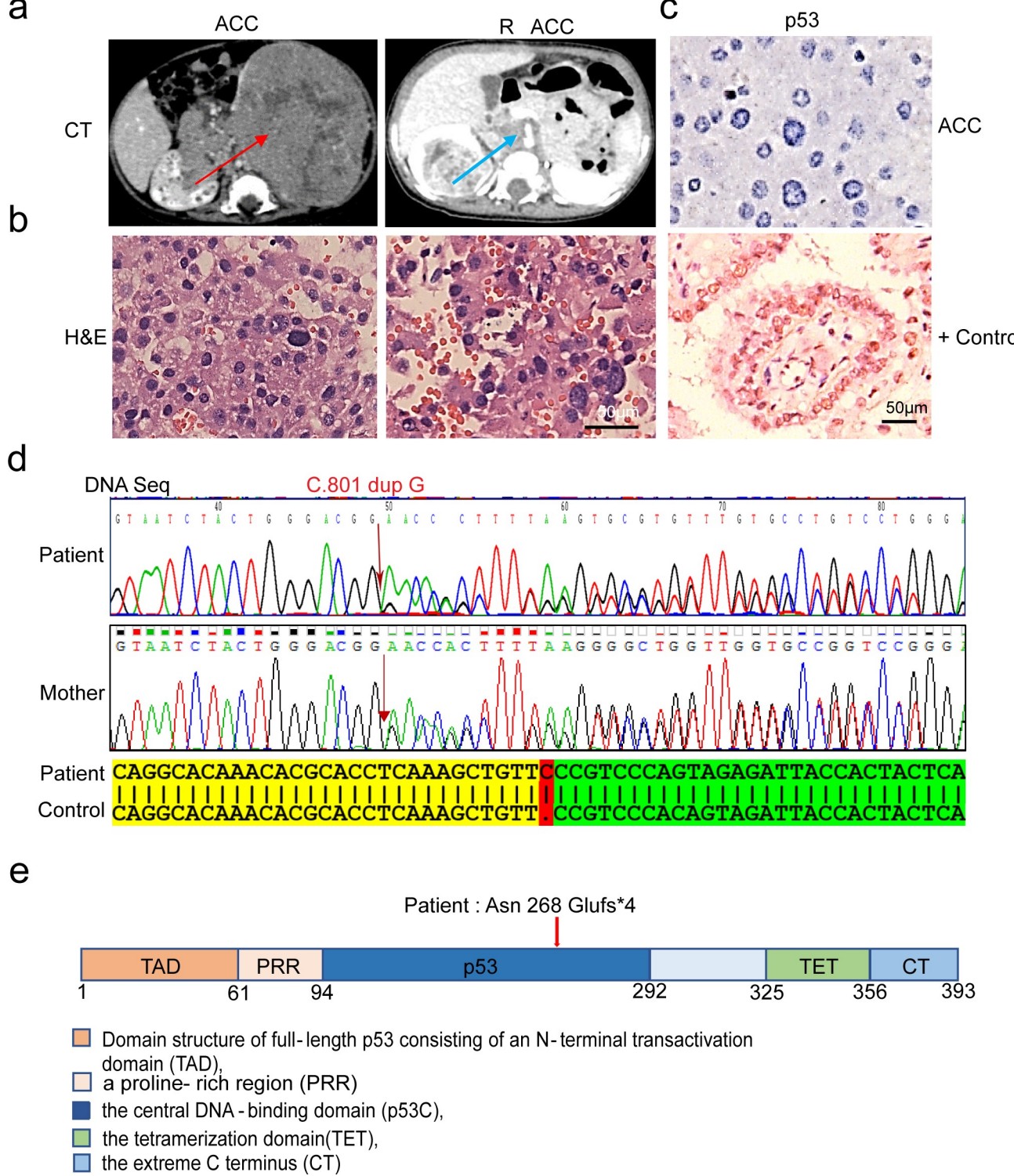

**Fig 1. Identification of a *Asn268Glufs\*4* mutation of *p53* in a LFS patient. a-b.** The patient was first diagnosed with composite ACC and neuroblastoma at the age of 6 months. Relapse of ACC was diagnosed when he was 16 months old. CT of the mass arising from the left adrenal gland at initial presentation (red arrow) and in the right adrenal gland at relapse (blue arrow). Histologic appearance (H&E staining) of the adrenocortical carcinoma at diagnosis and relapse stage. **c.** No expression of p53 in the left adrenocortical carcinoma cells from the patient. **d.** Sanger sequencing of the patient and his mother. The mutation site of *p53* is indicated by the red arrow. The *p53* sequence is C.801 dup G on chromosome 17 in the patient and his mother. **e** The domain structure of full-length p53 consisting of an N-terminal transactivation domain (TAD), followed by a proline-rich region

(PRR), a central DNA-binding domain (p53C), a tetramerization domain (TET), and an extreme C-terminus (CT)The *p53* mutant position of the patient is indicated by the red arrow.

Next, we generated iPSCs from the mononuclear cells of this patient, his mother and three healthy individuals, respectively. On the 14th day of reprogramming, we counted the number of iPS colonies. The number of iPS colonies from the patient was significantly less than his mother and the mean number of three healthy control (Fig 3B). The induction efficiency of the patient's iPS was also declined (Fig 3C). This data shows that this *p53* mutation gets a new function, which inhibits somatic cell reprogramming.

Since p53 is the best known 'guardian' of the genome and the loss of p53 function can induce the abnormal karyotype [28, 29], we randomly picked up three iPS cell lines and performed karyotype analysis. As shown in Fig 3D, all of the chromosome numbers of iPSCs were hypodiploidy. To confirm that *p53* mutant inhibited somatic reprogramming, we separately introduced *p53* mutant, WT, or an EV control into *p53*⁻/⁻ and *p53*⁺/⁺ mouse embryonic

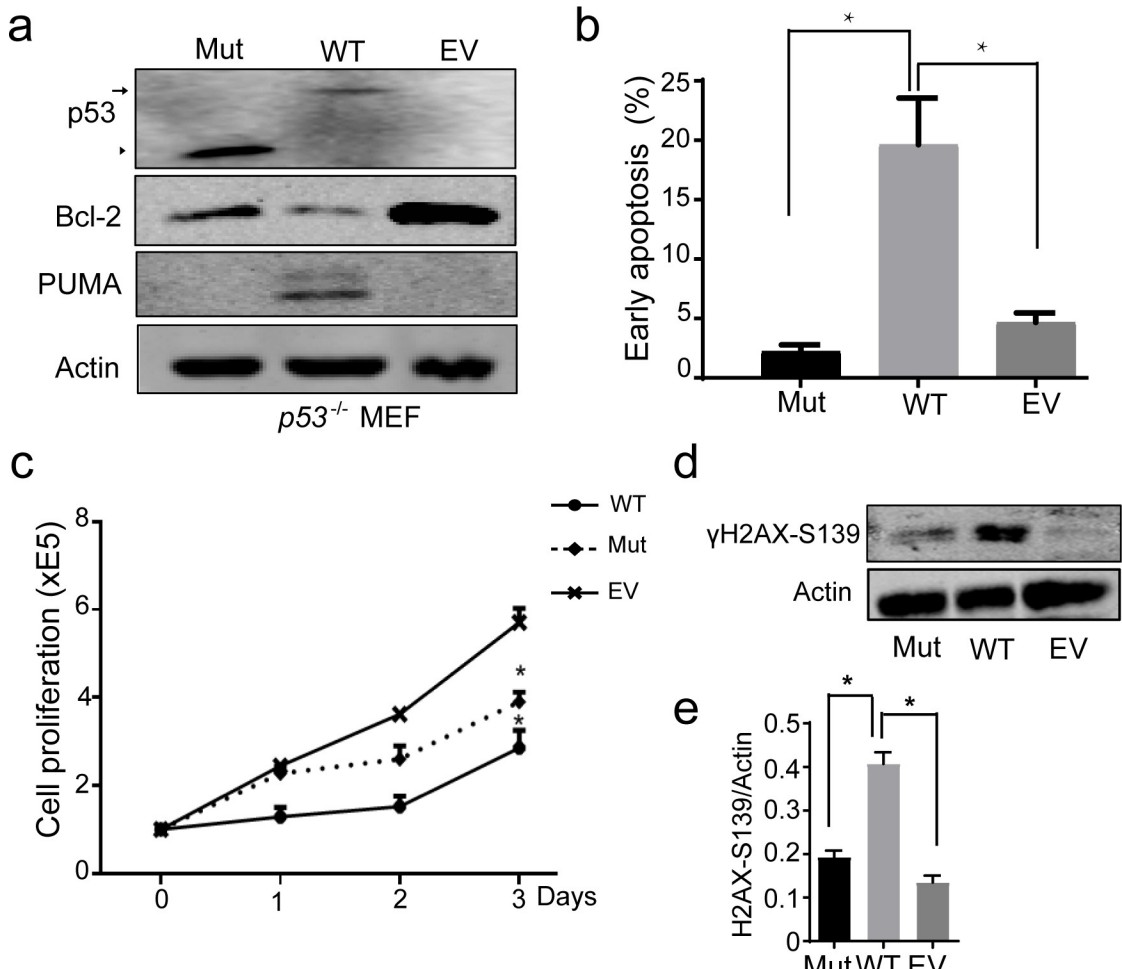

**Fig 2. p.Asn268Glufs\*4 mutation of p53 loses some functions of wild type p53. a.** Western blotting (WB) of expression of p53, BCL-2, and PUMA in *p53*⁻/⁻ MEF transfected with lentiviruses-mediated *p53* WT (WT), mutant (Mut), or an empty vector (EV) control. Arrow, WT p53; arrow head, p53 mutant. **b.** FACS analysis of apoptosis at Day 3 in the cells from **a**. * *p* < 0.05. **c.** Cell proliferation analysis. **d.** WB of γH2AX-139 expression. **e.** Quantitative analysis of γH2X-S139 protein expression in **d**.

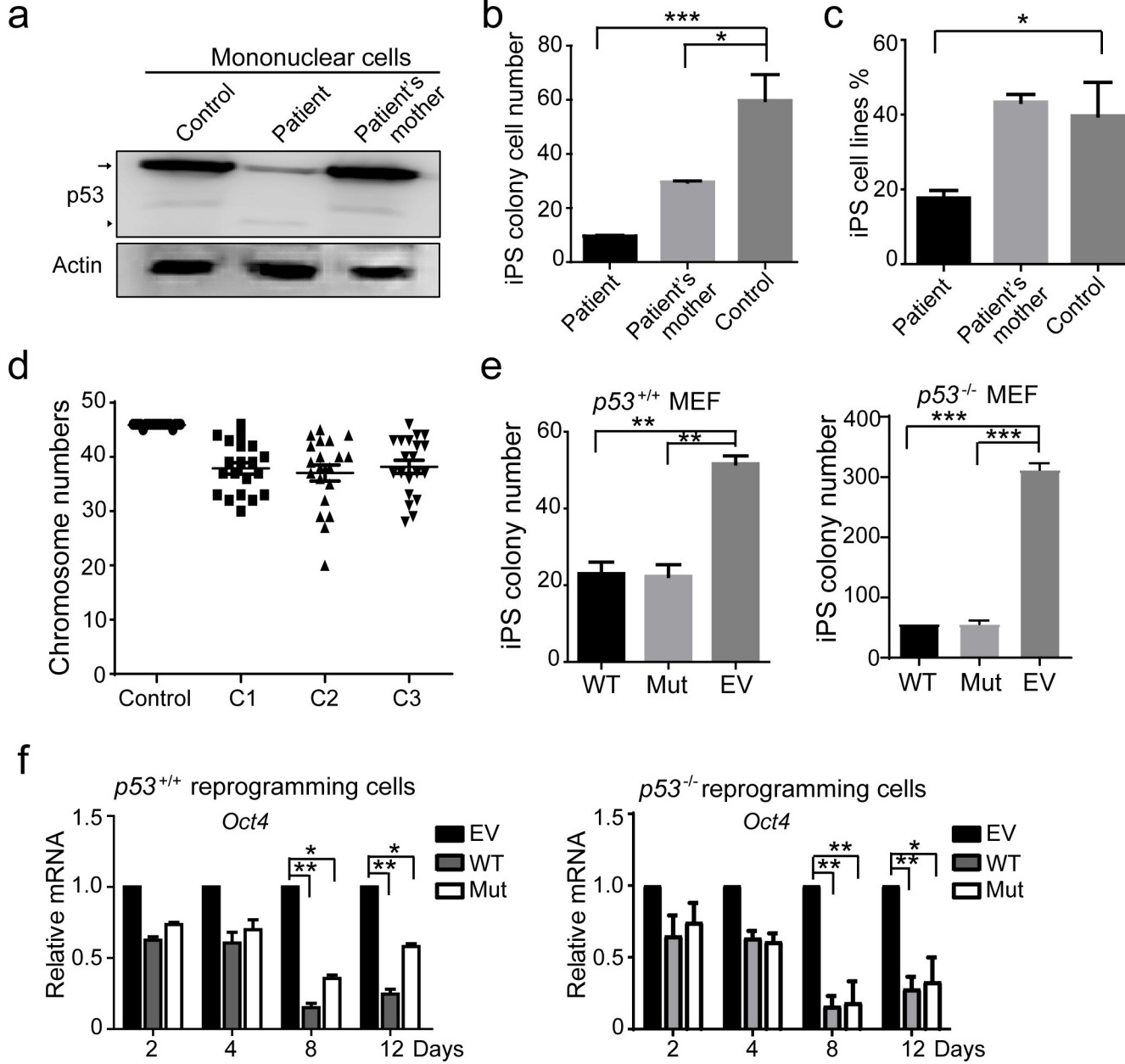

**Fig 3. The *p53* p.Asn268Glufs*4 mutation inhibits iPS cell generation. a.** WB of p53 protein levels. Mononuclear cells in healthy people with the same age as the patient were used as a control. Arrow, WT p53; arrow head, p53 mutant. **b.** iPS colony numbers per $1 \times 10^6$ monocyte cells used to generate iPSCs at Day 14 after transduction. ***, $p < 0.001$. **c.** The percentage of iPS cell lines established on Day 16 after transduction. **, $p < 0.01$. **d.** Chromosome numbers in three patient iPS cell lines. **e.** iPS colony numbers following introduction of WT p53, mutant p53 and vector into p53[+/+] and p53[-/-] MEFs were counted on reprogramming Day 14 after transduction. **f.** Real-time PCR (RT-PCR) of expression of OCT4 in cells following introduction of WT p53 and mutant p53 compared with vector control at the indicated reprogramming time points.

fibroblast (MEF) cells and then MEFs were reprogrammed to iPS cells. The numbers of iPS colonies in the mutant and WT groups on the 14th day of reprogramming were significantly lower than that in the control group (Fig 3E). However, p53 R175H did not affect the reprogramming rate (S1F Fig). Compared the expression of pluripotent genes on Day 2nd, 4th, 8th, and 12th during reprogramming, Oct4 expression in *p53* WT and mutant cells had been

significantly less than that in the control (Fig 3F), whereas the expression of SOX2 and NANOG had no difference (S1A and S1B Fig). This data indicates that the mutant of p53 likes as its WT and inhibits Oct4 expression and reduces the reprogramming efficiency.

To investigate whether *p53 p.Asn268Glufs*4* mutation influenced cell pluripotency, three *p53 p.Asn268Glufs*4* iPS cell lines were picked up. Using RT-PCR, we found that the expression of pluripotency genes, including OCT4, SOX-2, NANOG, and Rex-1 in these three *p53* mutant iPS cell lines was coincident with the H1 ESC at RNA level (S2A Fig). At protein levels, we also confirmed that all three iPSCs retained ES marker expression (e.g., Oct4, Sox2, NANOG and TRA-1-60, S2B Fig) by immunostaining. What's more, the iPSCs with the *p53 p. Asn268Glufs*4* mutation as normal iPS could differentiate into three primary germ layers and form teratomas in immunodeficient mice (S2C Fig). All of these data indicate that iPSCs with the *p53 p.Asn268Glufs*4* mutation can maintain pluripotency. Ultimately, similar to previous reports [30–32], we could not detect the vector sequence (EBNA1 and OSW) in iPSCs by PCR after 10 times of passages (S2D Fig).

## The heterozygous *p53* mutant cells have random allelic expression of p53

To demonstrate whether the iPS was originated from this patient, we performed Sanger sequence analysis. The results showed that all iPS cell lines contained the same *p53* mutation with patient's somatic cells. (Fig 4A), which confirmed that the *p53 p.Asn268Glufs*4* mutant is a germline mutation.

Clinically, it is common for LFS patients to carry the p53 mutations. However, not all p53 mutations carriers will develop into LFS patients p53 does not fully follow the classic Knudson's two-hit theory during carcinogenesis or cancer progression [33, 34]. Similar to the previous condition, the patient here inherited the disease-causing mutation, *p53 p.Asn268Glufs*4* from his mother, but his mother (22 years old) had not yet developed the disease. Compared with the expression level of p53 between different iPS cell lines from the patient, we found that there was no difference at their mRNA levels (Fig 4B) whereas their protein levels were significantly different (Fig 4C). p53 protein levels in one of the iPS cell lines were same as in H1 ESCs, whereas the other two iPS cell lines expressed lower levels of p53 WT and mutant proteins (Fig 4C). To clarify this phenomenon, we performed Sanger sequencing of the p53 cDNAs from the three patient iPS cell lines and found that the cell line with normal amount of p53 protein only contained the *p53* WT, while the other two with lower expression of p53 protein contained almost equivalent amounts of the WT and the mutated *p53* sequences (Fig 4D) Then, we checked p53 mRNA and protein levels in other three randomly selected iPS cell lines, but we did not find any difference compared to the H1 ES control (S3A and S3B Fig). What's more, all of the three iPS cell lines contained *p53* WT RNA sequence (S3C Fig). These data indicated that there may be random allelic gene expression in *p53* heterozygous mutations.

To confirm our hypothesis, we sequenced p53 cDNA from the patient's and his mother's mononuclear cells. We found that their mononuclear cells mainly contained the WT p53 sequence and low expression of the mutated p53 RNA (Fig 4E). It suggested that the *p53* mutant allele was expressed in iPS cell lines and somatic cells. This finding indicates that checking the protein level of mutant p53 may be more important than sequencing *p53* DNA and mutant *p53* allelic expression is a potential predictor of cancer risk.

## Discussion

In summary, the goal of the present study was to gain a better understanding of the specific roles of *p53* mutations during iPSC reprogramming and p53 related tumorigenesis. Mutations

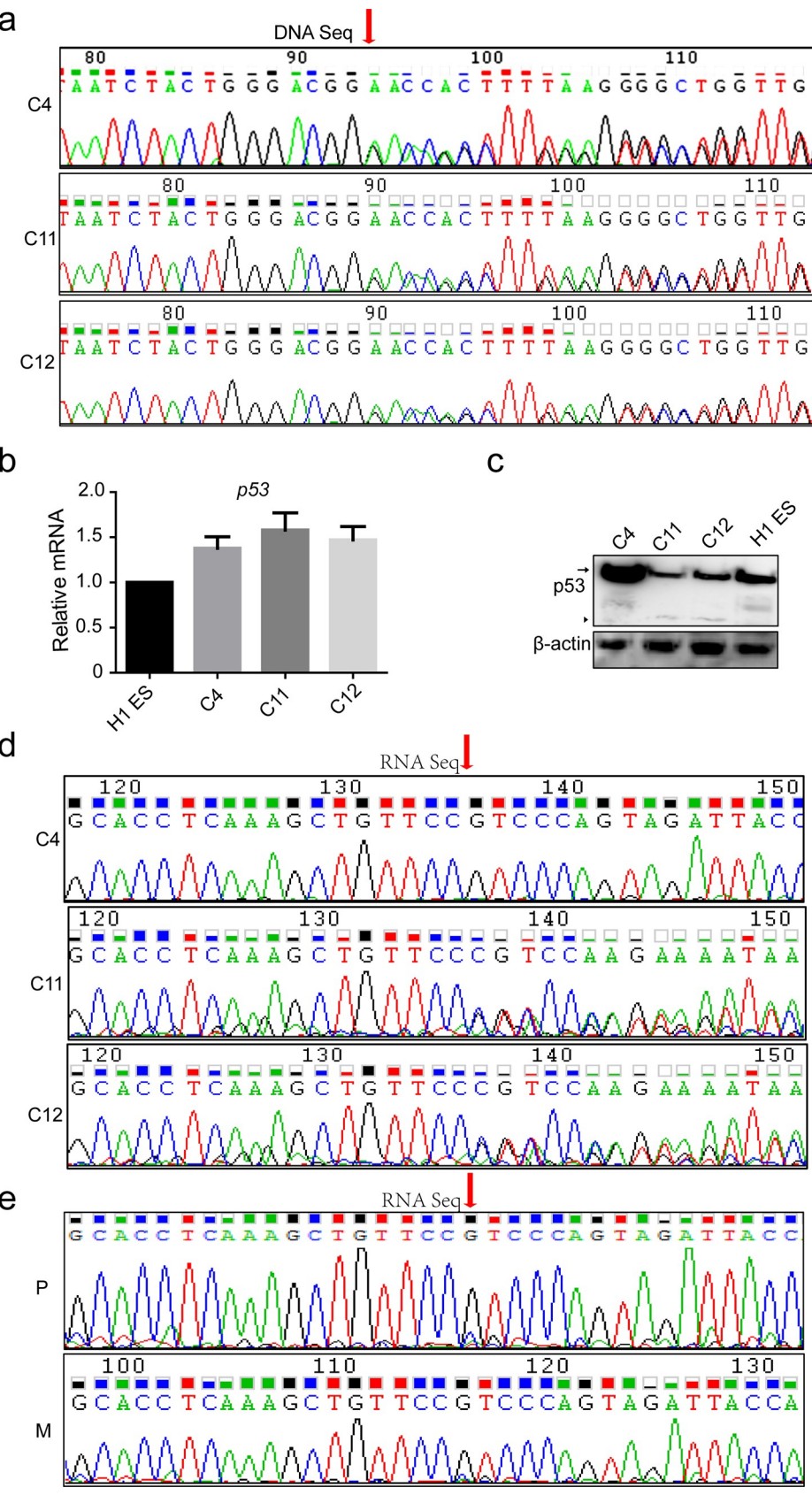

**Fig 4. The *p53* mutation causes random allelic expression in heterozygous iPS cell lines. a.** Sanger DNA sequencing of three patient iPS cell lines. $^{**}$, $p< 0.01$. $^{*}$, $p< 0.05$. **b.** RT-PCR of expression of *p53* in iPSCs derived from an LFS patient compared with H1 cells. **c.** WB of p53 protein expression in iPSCs derived from an LFS patient compared with H1 cells. Arrow, WT p53; arrow head, p53 mutant. **d.** p53 cDNA sequence from three LFS patient-derived iPS cell lines. **e.** p53 cDNA sequence from the somatic cells of the patient and his mother.

in *p53* usually not only abolish its normal function, but also gain additional oncogenic functions [13–16]. In a word, this specific *p53* mutation loses the ability to induce apoptosis and inhibit proliferation, which facilitates tumorigenesis. Besides, this mutation reduced the efficiency of somatic cell reprogramming by inhibiting OCT4 expression. Random allelic expression of p53 in heterozygous *p53* mutations caused variable WT p53 protein expression, which might be one of the reasons why people with the same *p53* mutation had different states of health.

In the present study, we found that losing part of p53 function caused by a heterozygous mutation did not promote cell reprogramming, instead, it did significantly decrease induction frequency of iPS generation by inhibiting OCT4 expression during reprogramming. This result was contrary to the phenomenon related with p53 deletions and mutations (p53 isoform *Δ133p53*). For example, the high expression of the *p53 isoform Δ133* improved the induction efficiency of iPSCs and ensured genomic integrity during reprogramming [35–37]. Consistent with our results, the OCT4 expression dramatically increased in *p53* knockout MEF cells compared with WT *p53* MEF cells [38]. Using chromosome counting, we found that three iPS cell lines were hypodiploidy, which was the same as that in the *p53* knockout iPS cell lines [12].

Family history could not predict the presence of an underlying predisposition syndrome in most patients [39]. In mammals, monoallelic gene expression can result from X-chromosome inactivation, genomic imprinting and random monoallelic expression (RMAE) [40, 41]. Recently, many studies have found allelic imbalance in the chromatin state of autosomal genes [42–44]. Biallelic inactivation of p53 has a significant impact on clinical outcome in multiple myeloma [41].

In our paper, we found that the patient and his mother had the same *p53* mutation, but his mother was a healthy carrier without any clinical symptoms. Using Sanger sequencing to analyze the p53 cDNA of the six patient-derived iPS cell lines, we found that four iPS cell lines only contained the *p53* WT cDNA sequence, while the other two with low p53 expression contained both WT and mutant *p53* cDNA sequences, indicating that *p53* random allelic expression occurred in heterozygous mutations. When testing the cDNA sequence of the patient and his mother's somatic cells, we also found very little mutant p53 RNA. This result confirmed that the random allelic expression of p53 in heterozygous *p53* mutations varied the WT p53 expression. Random allelic expression of heterozygous p53 mutations may be a reason why the people with p53 mutations develop cancer at random. This finding suggested that mutated *p53* allelic expression should be added to the risk forecasting of cancer.

## Conclusion

Our data demonstrate that the mutation of *p53 p.Asn268Glufs*$^*4$ maintains partial p53 function, which decreases the efficiency of somatic reprogramming by inhibiting OCT4 expression during the reprogramming stage and exhibites random *p53* allelic expression in heterozygous *p53* mutant cells. Random allelic expression of p53 in heterozygous mutation scenarios may be a reason why the people who carry p53 mutations develop cancer at random. Our finding also suggests that the mutant *p53* allelic expression may be a risk forecasting of cancers.

## Supporting information

**S1 Fig. Expression of pluripotent genes in *p53* mutation cells. a-b.** RT-PCR of expression of SOX2 and NANOG in cells with *p53* WT or mutant compared with an empty vector (EV) control. **c.** Western blot analysis of p53, BCL-2, and PUMA, γH2AX-139 expression after transfecting with lentiviruses carrying the p53 R175H, WT p53 and vector control plasmids into p53 KO MEF cells. **d.** Growth curve of p53 KO MEF cells with p53 WT or R175H. ** p<0.01. **e.** FACS analysis of apoptosis at day 3 after p53 KO MEF cells infection of p53 WT or R175H. ** p<0.01. **f.** iPS colony numbers following introduction of WT p53, R175H and vector into p53 KO MEFs were counted on reprogramming day 14 after transduction.
(PDF)

**S2 Fig. Qualification of iPSCs from LFS patient. a.** RT-PCR of expression of pluripotency genes in iPSCs compared with H1 ESCs. **b.** Representative images of pluripotency markers OCT4, SOX-2, NANOG, and TRA-1-60 in iPSCs. **c.** Teratoma analysis of iPSCs with *p53* mutation. H&E staining of representative teratoma with derivatives of three embryonic germ layers: blood vessel with blood (mesoderm), glands (endoderm), and epithelium (ectoderm). **d.** Vector sequence (OSW and EBNA1) was tested by PCR-based detection in iPSCs expanded for 10 passages.
(PDF)

**S3 Fig. Analysis of random allelic expression of p53 in another three iPS cell lines. a.** RT-PCR of expression of *p53* in another three iPS cell lines compared with H1 cells. **b.** WB of p53 protein levels in another three iPS cell lines compared with H1 cells. **c.** *p53* cDNA sequence from another three iPS cell lines.
(PDF)

**S1 Raw Images.**
(PDF)

**S1 Data.**
(DOCX)

## Author Contributions

**Conceptualization:** Yanxin Li.

**Data curation:** Yang Li, Ting Li, Yuejia Tang, Zhiyan Zhan, Lixia Ding, Lili Song, Tingting Yu, Yi Yang, Jing Ma, Yingwen Zhang, Ying Zhou, Song Gu, Min Xu, Yijin Gao.

**Funding acquisition:** Yanxin Li.

**Supervision:** Yanxin Li.

**Writing – original draft:** Yang Li, Ting Li.

**Writing – review & editing:** Yang Li.

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
