## [Decision Letter · Decision Letter 0]

6 Mar 2020

PONE-D-20-04840

The function of a heterozygous p53 mutation (p.Asn268Glufs*4) in a Li-Fraumeni syndrome (LFS) patient

PLOS ONE

Dear Dr Li,

Thank you for submitting your manuscript to PLOS ONE. After careful consideration, we feel that it has merit but does not fully meet PLOS ONE’s publication criteria as it currently stands. Therefore, we invite you to submit a revised version of the manuscript that addresses the points raised during the review process.

We would appreciate receiving your revised manuscript by Apr 20 2020 11:59PM. To enhance the reproducibility of your results, we recommend that if applicable you deposit your laboratory protocols in protocols.io, where a protocol can be assigned its own identifier (DOI) such that it can be cited independently in the future. For instructions see: http://journals.plos.org/plosone/s/submission-guidelines#loc-laboratory-protocols

We look forward to receiving your revised manuscript.

Kind regards,

Sumitra Deb, PhD

Academic Editor

PLOS ONE

Journal Requirements:

"This work was supported in part by the National Key R&D Program of China (2018YFC1313000/2018YFC1313005 Y. L and Y. G); the National Natural Science Foundation of China (No.81470315 and No. 81972341 to Y. L.); the Shanghai Jiao Tong University Medical Engineering Cross Fund (No. YG2017MS32); the Local High Level University Construction Project of Shanghai Jiao Tong University School of Medicine; the Commission of Shanghai Municipality (No. 17441903200 to S. G and No. 17411950402 to M. X.); and the Pudong New Area Science & Technology Development Fund (PKJ2018-Y47) to Y. L."

Reviewers' comments:

Reviewer's Responses to Questions

**Comments to the Author**

1. Is the manuscript technically sound, and do the data support the conclusions?

Reviewer #1: No

Reviewer #2: Partly

2. Has the statistical analysis been performed appropriately and rigorously? 

Reviewer #1: I Don't Know

Reviewer #2: Yes

3. Have the authors made all data underlying the findings in their manuscript fully available?

Reviewer #1: Yes

Reviewer #2: Yes

4. Is the manuscript presented in an intelligible fashion and written in standard English?

Reviewer #1: No

Reviewer #2: Yes

5. Review Comments to the Author

Reviewer #1: In this manuscript the authors have described a heterozygous p53 mutation in a LFS patient. The authors claim that p.Asn268Glufs*4 mutation promotes tumorigenecity and inhibit the efficiency of somatic reprogramming by inhibiting OCT4 expression. The manuscript has some major issues that need to be resolved.

1. Authors need to thoroughly proofread the manuscript as it had grammatical mistakes.

2. Authors need to be consistent with gene and protein nomenclature of p53 protein throughout the manuscript.

3. The sentence “P53 has four identical chains of 393 residues” in the result section is faulty syntax. Please revise.

4. What did the authors mean by “Asn268Glufs*4 mutation was located in specific DNA binding domain to stop this specific protein synthesis”. Does that mean that the mutation was a nonsense mutation and hence no protein was synthesized?

5. Authors need to clarify why they chose 293T cells that has endogenous WT p53 to do lentiviral infection experiments. The authors need to choose a cleaner system to do these studies with a p53 null background and then compare WT p53 vs mutant p53 effects.

6. In figure 2a why are there two different lanes for mutant vs WT p53? Also, the p53 blot is a little confusing.

7. Authors wrote that “The expression of PUMA in mutant 15 cells was the same as that in the control cells, but PUMA levels were relatively decreased in WT cells (Fig. 2a).” but that is contrary to what is shown in the figure. PUMA levels were increased.

8. Authors need to clarify between which groups was statistical comparison performed. Was it between control vs WT and control vs mutant or WT vs mutant? Was there any difference between control and mutant?

9. The result “TP53 p.Asn268Glufs*4 mutation promotes the tumorigenesis” cannot be concluded using just apoptosis and DNA damage experiments. Especially since most of the difference between mutant and control are minimal and may not be biologically significant.

10. How are the authors concluding that mutant p53 protein is present in the patient’s MNC? The levels that they are seeing in 3a-3b could be just the endogenous WT p53 allele since patient is heterozygous.

The mutation studied by the authors appears to be a loss of function mutation leading to the patient having only one functional WT allele. And most of the functional data shown could be attributed to having lower p53 levels due to only one allele present in the patient rather than the mutation specifically.

Reviewer #2: The manuscript titled “The function of a heterozygous p53 mutation (p.Asn268Glufs*4) in a Li-Fraumeni

syndrome (LFS) patient” by Li, et al. presents a follow-up analysis of a TP53 mutation they found in a Li Fraumeni Syndrome patient. They utilize cell culture models and sequencing to determine the functional aspect of the mutation. While the study of specific mutations found in LFS patients may be interesting, provided they yield some generalizable knowledge, there are some serious flaws throughout the manuscript that must be addressed which are detailed below.

1. The authors claim (pg. 14) that the mutation studied, Asn268Glufs*4, was located within the DNA binding domain of p53 which stopped protein synthesis. It isn’t mentioned anywhere in the rest of the paper that this mutation created a truncated protein, neither is it shown that a smaller p53 protein is expressed. Instead, the authors show that the mutant expresses at the same molecular weight as both wildtype p53 and control samples. (Figures 2A, 3B, 4C, and S3B)

2. The entirety of Figure 2 uses the 293T cell line, which has endogenous wildtype p53, to show the consequence of overexpressing the Asn268Glufs*4 mutant as well as wildtype p53 on wildtype p53 target proteins, apoptosis, cell proliferation, and DNA damage. In order to avoid the dominant negative effect of mutant p53 on wildtype p53, and to compare the mutant with wildtype p53, the authors should use a cell line without p53 for these studies.

3. Figure 2A-the “p53 mutation” panel-how is expression of the specific mutation probed for? Is this from a lower point in the gel? The antibody listed under Materials and Methods recognizes p53 at the N-terminus and thus would recognize both mutant and wildtype p53.

4. Figure 2A-the authors state (pg.15) that PUMA levels were decreased in WT cells, when in fact PUMA was significantly increased.

5. Figure 2C-the authors state that “this TP53 mutation can promote tumor cell proliferation”. The graph shown depicts less proliferation than control. In order to make this claim, the mutant should show greater proliferation than the control cells.

6. Figure 2D-densitometry should be shown to be able to make the claim that the mutant induced DNA damage.

7. The authors state at the end of the first paragraph on pg. 15 that the “mutant could promote tumorigenesis”. An actual tumorigenicity assay would need to be performed to make this claim.

8. Overall Figure 3-A comparison between two completely different individuals cannot be made because there is no control over any other genetic difference. An isogenic system would need to be used for comparison.

9. Figure 3A-what is “control”?

10. Figure 3B-if the patient is being compared to the mother, p53 expression needs to be shown in the same Western blot.

11. Figure 3F-what is the p53 status of the MEFs used? If the MEFs contain wildtype p53, then the same problem exists as in the 293T system for Figure 2.

12. Figure 4B-the text states that there was no difference between the patient’s and the patient’s mother’s p53 RNA level, but this is not shown.

13. Figure S2C-the authors state that the images come from tumors formed using immunodeficient mice. These need to be labeled-what are the authors trying to show? Also, do H1 ES cells form tumors themselves?

14. Another point-since the authors state that the mutation being studied here may have lost its function, it would be helpful to show a known functional mutant found in LFS patients as a positive control.

6. PLOS authors have the option to publish the peer review history of their article (what does this mean?). If published, this will include your full peer review and any attached files.

Reviewer #1: No

Reviewer #2: No

---

## [Author Response · Author response to Decision Letter 0]

6 May 2020

The due date for submitting the revised version of your article is 20 Apr 2020.

Point-by-point response to the reviewers’ comments:

Reviewer 1:

Q1. Authors need to thoroughly proofread the manuscript as it had grammatical mistakes.

Response: We thank the reviewer to point out this. We have carefully checked and corrected the full manuscript again.

Q2. Authors need to be consistent with gene and protein nomenclature of p53 protein throughout the manuscript.

Response: We agree with this comment. We have modified as p53 gene and p53 protein. 

Q3. The sentence “P53 has four identical chains of 393 residues” in the result section is faulty syntax. Please revise.

Response: We appreciate the reviewer to point out this mistake. We have corrected this as “The active p53 is a homo-tetramer formed by four identical chains of 393 residues each”. 

Q4. What did the authors mean by “Asn268Glufs*4 mutation was located in specific DNA binding domain to stop this specific protein synthesis”. Does that mean that the mutation was a nonsense mutation and hence no protein was synthesized?

Response: I apologize for this confusion. We have modified it to “Asn268Glufs*4 mutation is a nonsense mutation which is located in specific DNA binding domain, which caused early termination of this specific protein synthesis and may affect the function of p53.” 

Q5. Authors need to clarify why they chose 293T cells that has endogenous WT p53 to do lentiviral infection experiments. The authors need to choose a cleaner system to do these studies with a p53 null background and then compare WT p53 vs mutant p53 effects.

Response: We appreciate this important comment. We have carried out the experiment and replaced the data with p53-/- MEF (new Fig.2).

Q6. In figure 2a why are there two different lanes for mutant vs WT p53? Also, the p53 blot is a little confusing.

Response: We apologize for this confusion. We replaced this with new data showing p53 WT and mutant in the same film (new Fig. 2a).

Q7. Authors wrote that “The expression of PUMA in mutant 15 cells was the same as that in the control cells, but PUMA levels were relatively decreased in WT cells (Fig. 2a).” but that is contrary to what is shown in the figure. PUMA levels were increased.

Response: We apologize for these errors. We have corrected as: “PUMA levels relatively increased in WT cells”.

Q8. Authors need to clarify between which groups was statistical comparison performed. Was it between control vs WT and control vs mutant or WT vs mutant? Was there any difference between control and mutant?

Response: We apologize for this confusion. The statistical comparison was performed between control with p53 WT and p53 mutation. We have modified our result part to：“The analysis of apoptosis revealed that compared with unregulated control cells, overexpression of WT p53 enhanced apoptosis in p53-/- MEF cells (Fig. 2b, Fig.S1e). Only p53 WT induced DNA damage compared with the control (Fig. 2d-e, Fig. S1c). Unlike p53 R175H, p53 p.Asn268Glufs*4 mutant as well as its WT dramatically inhibited cell proliferation (Fig. 2c, Fig. S1d).”

Q9. The result “TP53 p.Asn268Glufs*4 mutation promotes the tumorigenesis” cannot be concluded using just apoptosis and DNA damage experiments. Especially since most of the difference between mutant and control are minimal and may not be biologically significant.

Response: We apologize for these errors. We have corrected as: “p.Asn268Glufs*4 mutation of p53 loses some functions of wild type p53”

Q10. How are the authors concluding that mutant p53 protein is present in the patient’s MNC? The levels that they are seeing in 3a-3b could be just the endogenous WT p53 allele since patient is heterozygous.

Response: We apologize for these errors. We replaced this with p53 WT and mutant band in the same film (new Fig. 3a).

Reviewer 2:

Q1. The authors claim (pg. 14) that the mutation studied, Asn268Glufs*4, was located within the DNA binding domain of p53 which stopped protein synthesis. It isn’t mentioned anywhere in the rest of the paper that this mutation created a truncated protein, neither is it shown that a smaller p53 protein is expressed. Instead, the authors show that the mutant expresses at the same molecular weight as both wildtype p53 and control samples. (Figures 2A, 3B, 4C, and S3B)

Response: We apologize for this confusion. We replaced this with p53 WT and mutant bands in the same film to new Figures 2A, 3A, 4C, and S3B

Q2. The entirety of Figure 2 uses the 293T cell line, which has endogenous wildtype p53, to show the consequence of overexpressing the Asn268Glufs*4 mutant as well as wildtype p53 on wildtype p53 target proteins, apoptosis, cell proliferation, and DNA damage. In order to avoid the dominant negative effect of mutant p53 on wildtype p53, and to compare the mutant with wildtype p53, the authors should use a cell line without p53 for these studies.

Response: We agree with this comment. We selected p53-/- MEF to repeat again. The results added to new Fig.2

Q3. Figure 2A-the “p53 mutation” panel-how is expression of the specific mutation probed for? Is this from a lower point in the gel? The antibody listed under Materials and Methods recognizes p53 at the N-terminus and thus would recognize both mutant and wildtype p53.

Response: I apologize for this confusion. We added the antibody to p53 information to material part: “Cell extracts were prepared, resolved on gels, transferred to nitrocellulose and incubated with antibodies against the N terminus of p53, which can recognize mutant and wild type of p53”. 

Q4. Figure 2A-the authors state (pg.15) that PUMA levels were decreased in WT cells, when in fact PUMA was significantly increased.

Response：We apologize for these errors. We have modified as: “PUMA levels relatively increased in WT cells”.

Q5. Figure 2C-the authors state that “this TP53 mutation can promote tumor cell proliferation”. The graph shown depicts less proliferation than control. In order to make this claim, the mutant should show greater proliferation than the control cells.

Response: We apologize for these errors. And we modified to: “Only p53 WT induced DNA damage compared with the control (Fig. 2d-e, Fig. S1c). Unlike p53 R175H, p53 p.Asn268Glufs*4 mutant as well as its WT dramatically inhibited cell proliferation (Fig. 2c, Fig. S1d)”.

Q6. Figure 2D-densitometry should be shown to be able to make the claim that the mutant induced DNA damage.

Response: We agree with this comment. We have added the densitometry data in new Fig.2.

Q7. The authors state at the end of the first paragraph on pg. 15 that the “mutant could promote tumorigenesis”. An actual tumorigenicity assay would need to be performed to make this claim.

Response: We agree with this comment. We have corrected as: “p.Asn268Glufs*4 mutation of p53 loses some functions of wild type p53”

Q8. Overall Figure 3-A comparison between two completely different individuals cannot be made because there is no control over any other genetic difference. An isogenic system would need to be used for comparison.

Response: We agree with this comment. We cannot get the isogenic system control, so we deleted this panel from Fig.3.

Q9. Figure 3A-what is “control”?

Response: We apologize for this confusion. The Control was normal MNC. Now we have deleted this panel from Fig. 3. 

Q10. Figure 3B-if the patient is being compared to the mother, p53 expression needs to be shown in the same Western blot.

Response: We agree with this comment. We have repeated this and showed p53 WT and mutant bands in the same film in new Fig.3a.

Q11. Figure 3F-what is the p53 status of the MEFs used? If the MEFs contain wildtype p53, then the same problem exists as in the 293T system for Figure 2.

Response: We agree with this comment. We added p53-/- MEF to do the experiments (Fig.3e-f, and Fig.S1a-S1b, S1f) again. The similar results were gained.

Q12. Figure 4B-the text states that there was no difference between the patient’s and the patient’s mother’s p53 RNA level, but this is not shown.

Response: We apologize for these errors. In Fig. 4B, iPS cells all come from patients. So, we modified the text states to “Compared with the expression level of p53 between different iPS cell lines from the patient, we found that there was no difference at their mRNA levels (Fig. 4b) whereas their protein levels were significantly different (Fig.4c).” in result part.

Q13. Figure S2C-the authors state that the images come from tumors formed using immunodeficient mice. These need to be labeled-what are the authors trying to show? Also, do H1 ES cells form tumors themselves?

Response: We apologize for this confusion. All iPS or ES which have the pluripotent ability can differentiate to three germ line tissues. So we modified to: “the iPSCs with the p53 p.Asn268Glufs*4 mutation as normal iPSCs could differentiate into three primary germ layers and form teratomas in immunodeficient mice (Fig. S2c). All of these data indicate that iPSCs with the p53 p.Asn268Glufs*4 mutation can maintain pluripotency”

Q14. Another point-since the authors state that the mutation being studied here may have lost its function, it would be helpful to show a known functional mutant found in LFS patients as a positive control.

Response: It’s a good suggestion. We have picked up p53 R175H as a positive control and performed new experiments. The result was shown in Fig.S1c-f.

---

## [Decision Letter · Decision Letter 1]

22 May 2020

The function of a heterozygous p53 mutation in a Li-Fraumeni syndrome patient

PONE-D-20-04840R1

Dear Dr. Li,

We are pleased to inform you that your manuscript has been judged scientifically suitable for publication and will be formally accepted for publication once it complies with all outstanding technical requirements.

With kind regards,

Sumitra Deb, PhD

Academic Editor

PLOS ONE

Additional Editor Comments (optional):

Reviewers' comments:

Reviewer's Responses to Questions

**Comments to the Author**

1. If the authors have adequately addressed your comments raised in a previous round of review and you feel that this manuscript is now acceptable for publication, you may indicate that here to bypass the “Comments to the Author” section, enter your conflict of interest statement in the “Confidential to Editor” section, and submit your "Accept" recommendation.

Reviewer #1: All comments have been addressed

Reviewer #2: All comments have been addressed

2. Is the manuscript technically sound, and do the data support the conclusions?

Reviewer #1: (No Response)

Reviewer #2: Yes

3. Has the statistical analysis been performed appropriately and rigorously? 

Reviewer #1: (No Response)

Reviewer #2: Yes

4. Have the authors made all data underlying the findings in their manuscript fully available?

Reviewer #1: (No Response)

Reviewer #2: (No Response)

5. Is the manuscript presented in an intelligible fashion and written in standard English?

Reviewer #1: (No Response)

Reviewer #2: Yes

6. Review Comments to the Author

Reviewer #1: There were still many grammatical issues during sentence construction. Please proof read thoroughly once more before it can be ready for publishing.

Reviewer #2: (No Response)

7. PLOS authors have the option to publish the peer review history of their article (what does this mean?). If published, this will include your full peer review and any attached files.

Reviewer #1: No

Reviewer #2: No

---

## [Editor Report · Acceptance letter]

27 May 2020

PONE-D-20-04840R1 

The function of a heterozygous p53 mutation in a Li-Fraumeni syndrome patient 

Dear Dr. Li:

I am pleased to inform you that your manuscript has been deemed suitable for publication in PLOS ONE. Congratulations! Your manuscript is now with our production department. 

With kind regards,

on behalf of

Dr. Sumitra Deb 

Academic Editor

PLOS ONE